# Boosting a Weather Monitoring System in Low Income Economies Using Open and Non-Conventional Systems: Data Quality Analysis

**DOI:** 10.3390/s19051185

**Published:** 2019-03-08

**Authors:** Daniele Strigaro, Massimiliano Cannata, Milan Antonovic

**Affiliations:** Institute of earth Sciences (IST), Department for Environment Constructions and Design (DACD), University of Applied Sciences and Arts of Southern Switzerland (SUPSI), CH-6952 Canobbio, Switzerland; massimiliano.cannata@supsi.ch (M.C.); milan.antonovic@supsi.ch (M.A.)

**Keywords:** 4onse, big data, Monitoring, istSOS, low-cost, OGC, SOS

## Abstract

In low-income and developing countries, inadequate weather monitoring systems adversely affect the capacity of managing natural resources and related risks. Low-cost and IoT devices combined with a large diffusion of mobile connection and open technologies offer a possible solution to this problem. This research quantitatively evaluates the data quality of a non-conventional, low-cost and fully open system. The proposed novel solution was tested for a duration of 8 months, and the collected observations were compared with a nearby authoritative weather station. The experimental weather station is based in Arduino and transmits data through the 2G General Packet Radio Service (GPRS) to the istSOS which is a software to set-up a web service to collect, share and manage observations from sensor networks using the Sensor Observation Service (SOS) standard of the Open Geospatial Consortium (OGC). The results demonstrated that this accessible solution produces data of appropriate quality for natural resource and risk management.

## 1. Introduction

This paper addresses the issue of the low resilience and adaptation capacity of low-income economies due to inadequate weather monitoring systems [1]. To this end, the authors evaluated the potential of open and non-conventional solutions to tackle existing barriers: high costs related to conventional solutions; low accessibility to the hardware and software components; missing data on interoperability; closed source of data format, hardware and software solutions.

Although several open hardware/software systems are presented in the literature [2,3,4,5,6], there is a need to prove its operative capacity and sustainability in real-case applications. This research presents a partial result of the 4onse project (www.4onse.ch) that aims to validate the technology in a real-case scenario at a watershed level to verify the effective sustainability of this system. To understand if this kind of solution can effectively be an alternative to a more expensive and conventional system, this paper presents a data quality analysis, rarely found in the literature, of a fully open prototype. This is a pre-requisite for further testing on a larger scale. The next steps of the research will evaluate the durability and sustainability of such a solution from a socio-economic perspective.

### 1.1. State of the Art of Monitoring in Developing Countries

Nowadays, the availability of large information datasets and powerful computing devices opens new perspectives. The Internet of Things (IoT) [7,8,9] is connecting more and more devices to the Internet, generating a massive amount of observations [10,11]. Thanks to these technologies, low-cost and non-conventional sensors can play an important role in environmental monitoring, either as a complement to standard and authoritative systems or as a vital data source in regions where traditional observation networks are in decline or missing [12]. In fact, these “crowdsourced” data [13] are very attractive due to being generally available in near-real time with a high spatio-temporal resolution and, therefore, potentially able to fill the long-standing gap in detecting local activities that go unnoticed in various geographic locations [14].

Nevertheless, Muller et al. [15] noted that despite the great potential of non-conventional data sources, before they can be fully exploited, a number of challenges need to be addressed. Data quality is a primary issue. Decisions which affect lives, properties or society shall be based on data with a sufficient level of reliability and accuracy. A second issue, linked with to the previous one, is the unavailability of metadata, such as, for example, maintenance log-books, equipment specifications, or position and deployment procedures. They include essential information for assessing the data reliability and accuracy produced by a specific source. A third issue is the standardization of data and metadata in order to build robust and widely applicable systems. Historically, manufacturers created vertical solutions that make use of their own technologies and closed services. Thus, standards are needed to change this “Intranet of Things” into the “Internet of Things” [11,16].

According to the United Nations [17], although several efforts have been recognized, critical data for effective policy making are still missing in large areas. The major challenges are the poor quality of the data, lack of timely observations and scarce availability of disaggregated information. Focusing on climate (which in a broad sense includes meteorology, oceanography and climatology) and developing countries, Snow [18] registered a lack of weather observing networks that routinely provide data on the necessary quality and spatio-temporal density. Some countries do not have monitoring networks, the existing ones are in decline, they are not connected in real time, and/or they are not dense enough to monitor local conditions that can evolve rapidly. The common identified causes are: limited (often very limited) budgets, the lack of technical infrastructure and associated expertise, rapid deterioration, lack of skilled maintenance, and high-priced non-locally available spare parts.

### 1.2. Opportunities, Assumptions and Open Questions

The present research is based on the fact that Open Source solutions adoption is leading the IT development [19,20], while fast growing mobile connectivity and smart device diffusion has been observed in developing countries [21,22]. The combined accessibility to communication, hardware and software could be a great opportunity for setting up monitoring systems at a low-cost, which could address a number of practical issues including, but not limited to, floods, smart agriculture, climate change, risk management, nowcasting and weather predictions.

However, the mentioned vision assumes a few sounding assumptions and hypotheses. The first commonly accepted assumption is that the IoT has a great potential for societal, environmental and economic impacts [9,16], enabling smarter decisions. Another supposition is that the lack of efficient, dense and modern monitoring systems in developing countries is due to the scarce sustainable solutions that meet local needs. This is supported by studies that identified, as a major reason for monitoring system failures in developing countries, the dependency on the overly costly replacement components and the missing local support capacity [18]. A third hypothesis is that in the coming years, the technological diffusion of mobile internet would reach most of the world, including not only the most populated areas but also the most remote regions. The evolution of this technology in developing countries cannot exactly be predicted, but there are marketing and economic studies that forecast a fast growing diffusion of wireless connectivity and smart devices in the short future [22]. Considering the state-of-the-art and the formulated hypotheses, a “fully Open” and “non-conventional” System for monitoring the environment (hereafter referred to as 4ONSE) may represent a possible response for globally solving the present issues in developing countries. For “fully Open”, the authors intend a system composed of Open Hardware, Software, Standards and Data, while for “non-conventional” they intend a low-cost system which does not respect all the high standard requirements in terms of the sensor construction, precision and testing.

The main open question is: could a 4ONSE system be a viable solution to monitor local phenomena in a sustainable and “effective way”, so as to meet the users’ needs? Unfortunately, despite the existence of research that investigated Open Hardware solutions [23,24] or Open Standard [25,26], to the knowledge of the authors, the combination of these two systems with open software and data has been investigated in only a few cases [27,28,29]. The question, which remains therefore largely under-explored, is partially addressed in this research which evaluates the quality of the 4ONSE system, and it is presented in the next sections. In particular, this paper discusses the system design and data quality analysis. Future research will investigate the durability and applicability of such a system deployed in real case experiments on a watershed scale.

### 1.3. Design and Requirement

As reported by Hill [30], the general Environmental Monitoring Systems (EMS) aims at collecting sensor readings from different locations in order to detect trends and interrelations. Most of the applications are designed to collect data to extract trends and forecasts along a period. The observed series should be acquired at regular steps and at fixed locations in time by a large number of stations. A continuous and reliable transmission permits the archival of data in a central data warehouse [28]. Long time series are mostly desired, while a long network lifetime is required. The sensors are, on average, evenly located in space, but specific hot-points may be additionally monitored. Depending on the desired size of the monitoring target area and the sensor density that is desired, it is possible to pre-calculate and implement transmission strategies that minimize the costs and data transmission failures. This can be implemented using gateways (interface between the application platform and the wireless nodes), and relay-nodes or “routers” (nodes used to extend the network coverage acting as a bridge to the gateway), which are connected to leaf-nodes (a physical interface between the network and the sensors/actuators). In weather applications, it is not necessary to have a dynamic routing strategy, since stations are located in a fixed position and then never moved. Specifically, for weather monitoring systems, the reporting period is between 10 and 30 min, since weather variables such as temperature, humidity, wind and rainfall do not change so quickly as to need higher reporting rates. A precise synchronization of the cycles of sleep mode and transmission are important to save energy and meet the long lifetime requirement [31]. Strict latency limits are not necessary since moderate delays in a sample transmission do not significantly affect the applications. The nodes are expected to occasionally fail; thus, periodic maintenance operations are essential to reconfigure or replace parts of the system. The self-energy sustaining of nodes should be guaranteed at the maximum. In the case of sustainable energy, it should guarantee the power during 5 days of unfavorable conditions (i.e. cloud days for solar panels).

In summary, the most important characteristics of the envisioned network that should be met are: a long lifetime, precise synchronization, low data rates, relatively static topology, and moderate data delay admitted. In addition to the above-listed characteristics, the 4ONSE project is expected to be: based on Open Hardware and Open Software; composed by non-conventional and low-cost sensing technologies; compliant with international Open Standards; and capable of handling a massive dataset and produce Open Data summary reports.

These four open pillars drowned all the design choices and constituted the mainstream of the design process. Additionally, the system should guarantee a good data quality and high quality of services to suit applications like the management of artificial basins or floods, and droughts risk reduction. Other desirable characteristics of the system are: a modular solution to better fit different needs, ability to support different sensors, alternative data collection methods (automatic or manual), different level of installation protection, and ease of use and maintenance.

The selected meteorological variables are: air temperature relative humidity, barometric pressure, light, wind speed and direction, and rainfall. The battery level and internal temperature can eventually be measured.

## 2. Materials and Methods

To test the data quality, the prototype has been tested against a reference station, based on a low-cost and open technologies, over eight months. This period was essential to evaluate if the 4ONSE solution fit the individuated needs. This section describes the experiment details in terms of the selected technologies, system architecture, implemented solution specifications and experiment setup and characteristics.

### 2.1. General System Technologies

To fulfill the requirements of using only open technologies, we selected: Arduino as the main element of the hardware component to get and transmit the measurements; Sensor Observation Service (SOS) from the Open Geospatial Consortium (OGC) as the open standard to handle the observations along the system in an interoperable way; the software istSOS (www.istsos.org) for managing the observations on the server side in ‘near-real-time’; Observation Analysis Tool (OAT) [32] as the Python library to elaborate and analyse data, and to produce summary reports; CKAN as open data portal to share statistical reports.

Among the available software implementation distributed with open source licenses, the istSOS (Istituto Scienze della Terra Sensor Observation Service) has been selected due to its specific features designed to handle the data management chain of hydro-climatic data and because it is totally compliant with the SOS OGC standard. In fact, it is currently used as data management platform to monitor the Verbano lake and to collect hydro-meteorological data from the Canton Ticino [33].

OAT is a Python library that has been recently developed within the H2020 FREEWAT project (www.freewat.eu). The library offers convenient methods to create a time-series object from different data sources (including istSOS), to apply processing algorithms, and to export results in a number of different formats. The library, which is easily extensible, offers the opportunity to implement a number of processes that permit complex a data validation and analysis, and to save outputs to an open data repository. Due to its large diffusion and the availability of Python action API, the software CKAN (Comprehensive Knowledge Archive Network; http://ckan.org/) will be used in the next steps of the project as an interface to enhance the access to data, metadata and reports. CKAN is a software solution for open data management and dissemination. It provides modules for spatial indexing, data preview, and service interfaces such as the OpenGIS Catalogue Service from OGC, and it allows users to find collections of scientific data quickly and easily.

### 2.2. General System Architecture

The architecture of the system is constituted by three layers: the hardware, service and communication layer. The hardware layer is composed of the weather station which provides observations about the selected environmental parameters, including the water river level. The river gauge will be used in the future steps of the project for hydrological models and studies. The service layer collects and archives the data which are transmitted to the management system istSOS following the standard OGC SOS specifications. Finally, the communication layer is the contact point between the hardware and the service layers. A fast communication method together with a strategy to prevent data loss was implemented and evaluated, in order to offer a robust and consistent data transmission through the 2G GPRS network, which is mostly found in developing countries.

### 2.3. The 4onse-Mod Solution

A solution, named 4onse-mod, has been specifically designed to maximize the reproducibility of the system by using common materials available worldwide (devices, box, screws, etc.) as much as possible. It is based on a Plexiglas board that hosts the power regulation controls on the back side, and all the other components on the front side (Figure 1). This choice facilitates the sensor replacement and prevents the user interaction with the power cables.

The weather station modules can be grouped in internal and external components (Figure 2). The internal components are: (i) the microcontroller board (Arduino MEGA 2560, www.arduino.cc), which is the core of the system. It manages the communication, it reads values from the sensors, and it logs them into the data logger; (ii) the SIM800 GPRS module (www. simcom.ee) to transmit data to the server; (iii) the DS3231 RTC (www.maximintegrated.com) to get the date and time values; (iv) the OpenLog data logger interface (www.sparkfun.com) to keep track of the sensor data; (v) the internal temperature sensor DHT11 (www.mouser.com), to monitor the environmental conditions inside the box; (vi) the step-down converters to reduce the voltage from 12 V to 7 V and to 5 V.

The external components are listed in Table 1. They are the sensors used to measure the air temperature, air pressure, air humidity, soil humidity and precipitation, as well as the wind speed and direction.

The station is powered with a 12 V input current through a solar or wired system. The voltage is then stepped down to 7 V to provide the right current to the Arduino board and then to 5 V to power all the other components.

The general installation schema of the 4onse-mod solution is illustrated in Figure 3. A central pole sustains the box container, where the unit controller is installed, along with the sensors that, due to best practice guidelines (https://www.wunderground.com/weatherstation/installationguide.asp), must be placed at an appropriate distance from the ground, and from surrounding buildings or other obstacles.

Since collecting continuous time series is essential in system monitoring, the 4onse-mod stations implements an efficient strategy to prevent data loss due to system failures. All the measured data are stored on an Secure Digital (SD) card, but the transmitted data are staged in a different area from the observations that are yet to be sent to the server. In case of a communication failure, the system is able to automatically recover non-transmitted data to fill the data gaps.

### 2.4. Experiment Setup and Characteristics

The data quality and robustness of the 4ONSE monitoring system is evaluated by comparing the measurements observed from the 4onse-mod station with an authoritative reference station of the hydro-meteorological network of Canton Ticino (Southern Switzerland), named Trevano, over a period of 8 months. The 4ONSE prototype was installed near the Trevano station (Figure 4) in order to observe the same microclimate. The Trevano weather station has been collecting data since November 3rd of 2007, measuring the following parameters: precipitation with 0.2 mm resolution; temperature with ±0.3 °C at a 25 °C accuracy, air relative humidity with ±4% over 0 to 100%; atmospheric pressure with ±0.5 mb at a 20 °C accuracy. As well as the 4onse-mod prototype, this reference station sends data to an istSOS instance that, thanks to the SOS standard format, facilitates the data retrieval for elaborations and analyses.

The accuracies and resolution of the 4onse-mod sensors available on the respective sensors datasheets are reported and listed as follows: the waterproof temperature sensor (DS18B20) has an accuracy of ±0.5 °C at 25 °C; the BME280 (www.bosch-sensortec.com) provides air pressure and air relative humidity data respectively with ±1 hPa at +25–40 °C and ±3% at 25 °C; and the rain gauge Davis Aerocone 6465 (www.davisinstruments.com) has a resolution of 0.2 mm.

The Davis pluviometer is the most expensive sensor used to develop the prototype, and it is a proprietary solution. During the project activities, we designed and implemented a 3D printed rain gauge which has been calibrated and is under testing. In future, a comparison analysis will be performed to prove the applicability of the proposed open solution. 3D models and the methodology to calibrate the rain gauge will also be made accessible according to the open philosophy of the project.

The period considered in this validation ranges from July 1st, 2017 to the February 28th, 2018. 4onse-mod was set with a data reading frequency of 5 min and a data transmission frequency of 15 min. Trevano collects data every 10 min and transmits them every 30 min.

Although the 4onse-mod station has an anemometer, an anemoscope and a soil humidity sensor, it was not possible to evaluate the quality of these data since Trevano does not have equivalent sensors. In both stations the air temperature sensor is placed in a solar radiation shield, respectively located at 1.8 (4onse-mod) and 2.2 (Trevano) meters above the ground. In agreement with the World Meteorological Organization (WMO) requirements [34], the air pressure and air humidity sensors are also mounted within the same shield.

The data quality was evaluated by comparing 10 min mean, daily minimum, daily maximum and daily mean values for each monitored parameter at the two stations. The comparison was performed by means of:
a visual comparison of the recorded data;a time series goodness-of-fit calculation through the coefficient of determination by Pearson;the coherence of aggregated values (daily mean, max and min values);scatter plots to evaluate the general behavior of the 4onse weather sensors;a probability density function to validate the residuals distribution.


## 3. Results and Discussion

This section presents the results of the experiment by discussing the different observed properties.

### 3.1. Air Temperature

In Table 2, the coefficient of determination by Pearson is calculated by comparing the time series of the reference and the 4onse-mod stations (Figure 5). The graph *a* shows the 4onse-mod and Trevano time series with the values aggregated and averaged every ten minutes. The *b*, the *c* and the *d* refer respectively to the daily maximum, minimum and mean temperature data. According to Table 2, the R-squared coefficients are very high for each of the compared data series, which means a very high relationship between the two stations’ trends. While R-squared provides an estimation of the strength of the relationship, the analysis of the residuals between the time series gives better evidences of the biases in the data collected. An in-depth evaluation was performed through the calculation of scatter plots based on the same data described above (Figure 6). Looking at the scatter plot *a* (10 min aggregated value), the high correlation between the temperature of the two stations is particularly clear. Almost all of the data falls inside an interval of −1 °C and +2 °C. The other three scatter plots help us find the goodness of the sensor in detecting the minimum, maximum and mean temperatures: the minimum temperatures are systematically overestimated, with a mean error of +0.26 °C (Table 2); the maximum temperatures are generally overestimated, and a greater variability is present compared to the minimum daily temperature; finally, the mean temperatures are totally comprised between the limits of 0 and +1 °C. This confirms the general behavior of overestimating the values.

In Figure 7, the density probability plots offer an overview on the deviations distribution. 90% of the residuals fall inside a range between −0.047 and 0.57 °C for the 10 min aggregated values; 0.08 and 0.45 °C for the daily minimum temperature values; −0.22 and 0.56 °C for the daily maximum temperature values; and 0.09 and 0.41 °C for the mean daily temperature values. The mean error values fit the mode of the curve in all the considered plots, except for the daily mean temperature density plot where two peaks can be identified: one with a lower probability than the mean value and the other, which corresponds to the most frequent error, with a higher probability than the mean value.

Finally, in the graphs in Figure 8, the trend of the residuals was analyzed using the daily mean temperature time series of the two stations. This analysis identified how, at lower temperatures (winter), the residuals increase and a lower accuracy is therefore expected. The residuals are near zero when high temperature values are recorded. Nevertheless, the observed residuals are always below 0.5 °C of variation and below 0.1 °C when temperatures are in close to 30 °C.

### 3.2. Air Relative Humidity

The visual comparison of the time series plots (see Figure 9) suggests that the 4onse-mod station overestimates high values and underestimates low values of humidity. Nevertheless, despite this behavior, the coefficients of determination, as reported in Table 3, are above 0.98. In fact, only the mean daily humidity time series has a value under 0.99, with an R-squared of 0.98, which is still very good.

The tendency to over/under-estimate values is better identified by the residual analysis of the 10 min aggregated values, which shows (see Figure 10) a non-linear trend with a “breakeven” point located at around 70%. This is confirmed also by the residual plots of the daily mean, minimum and maximum. As reported in Table 3, the mean error is higher for the daily average values (−3.87%), while the standard deviation is higher for the 10 min aggregated values (3.84%). As a consequence, the probability density plots (Figure 11) shows a non-negligible data dispersion around the mean with 95% of the values with residuals within ±7.7%.

Compared to the temperature probability density functions, the relative humidity time series curves (Figure 11) have a greater value dispersion since the probability peaks are not as high as the temperature analysis showed. In addition, the tails of the curves span over bigger intervals. Even if the results are good enough and acceptable, the measures of the relative humidity are not as accurate as for the previous parameter.

According to Figure 12, as was already noticed for the air temperature values, an increased value of the deviations is present in the winter observations. From the plots of the residuals vs. the mean temperature, it is possible to identify that: 1) the dispersion of the residuals is higher when lower values of air temperature are recorded; and 2) the residuals tend toward negative values with the increase of air temperatures.

### 3.3. Air Pressure

The pressure is important for understanding the dynamics of the atmosphere. According to the WMO definition [34], the atmospheric pressure on a given surface is the force per unit area exerted by virtue of the weight of the atmosphere above.

Both of the stations use hPa as a unit of measure, but they have different resolutions. Even though the 4onse-mod prototype gives the pressure as a double precision number, for the purpose of the analysis it was rounded to zero decimals since the Trevano weather station unfortunately provides the pressure as integer values.

The visual comparison of the time-series in Figure 13 suggests a very high correlation. This is confirmed by the coefficients of determination reported in Table 4. The two series follows the same trend with limited gaps in the correspondence of some minimum peaks. The scatter plot of the daily mean values shows (Figure 14) that the pressure data are on the identity line or only slightly overestimated. As shown in the probability density plots (Figure 15), 95% of the errors (2σ) of the ten minutes averaged, maximum, mean and minimum daily series span, respectively, over intervals of 0 and 1 hPa, −0.1 and 0.9 hPa, 0.1 and 0.6 hPa, and −0.2 and 1 hPa. The mean error is estimated at +0.4 hPa. Figure 16 shows the plot between the two time series of the daily mean pressure against the trend of the residuals. As for the air relative humidity and the air temperature parameters that were analyzed in the previous paragraphs, the pressure sensor has a greater drift in the residuals when lower temperatures are registered. The second plot in Figure 16 confirms this dependency of the pressure trend to the air temperature state. The lower the temperature, the higher the value of the variation with respect to the reference station.

### 3.4. Precipitation

From the analysis of the rainfall time series (Figure 17), it was possible to identify rainfall events that had occurred by using the *HydroEvents* feature available in OAT [32] that identifies an event by specifying a minimum number of hours between peaks, and a minimum value (rise_lag = 0.2 day, fall_lag = 0.2 day) to consider a peak (min_peak = 1.0 mm). During the testing period, 14 rainfall events were identified from both stations (Table 5). According to this table and to the scatter plot in Figure 17, the identified events show a lower deviation when the precipitation is below 20 mm. The error increases with growing values of precipitation: the event number 2 is overestimated by 4.8 mm; the events 3, 6, 7 are similar; the events 1, 2, 4, 5, 8 and 9 underestimate the precipitation events by more than 2 mm.

In Table 6 and Table 7, the maximum cumulated rain using two temporal moving windows of 24 h and 1 h is calculated. In this case, the sensor has a specular behavior: the maximum value of rain in 24 h is overestimated by +5 mm; the maximum value of the cumulated rain in 1 h is underestimated by −7 mm. The total amount of rain collected by the prototype is 632.4 mm, which differs from the reference by only 40.4 mm. During the testing phase, heavy rain events in summer and some snow events in winter affected the measurements.

## 4. Discussion and Conclusions

The United Nation [35] states: “Technological change, particularly in developing countries, is not only about innovating at the frontier, but also about adapting existing products and processes to achieve higher levels of productivity as applicable to their local contexts. In this process, the ability of local firms and enterprises to access technological know-how is fundamental to shaping their ability to provide products and services, which we believe are essential to improve living standards, and that could also promote growth and competitiveness.” In this sense, the 4ONSE monitoring system could be a great opportunity for developing countries to set-up monitoring systems with accessible technology at a low-cost, which could be used in future to address a number of practical issues including, but not limited to, flood-water and urban drainage management, climate change impact assessment, early warning and risk management, as well as now-casting and weather predictions.

If scientifically verified and validated, such a fully accessible, royalty-free and low cost system could potentially enable developing countries to monitor local phenomena in real-time and through dense (in time and space) observations. This could also lead to a better reaction time, understanding, wiser decision-making and effective policy implementations. Moreover, 4ONSE could also provide lower income countries with a fully accessible technology for the so-called “Internet of Things” economy. Based on the above considerations, the authors believe that, if the sustainability of the solution is proven and the quality of the data is verified to be enough for practical application, its integration into decision-making processes and policies will most likely be supported by governments and stakeholders. With this research, the authors have implemented a non-conventional solution based on open technologies and low-cost components and tested its data quality. The 4onse-mod prototype was evaluated in terms of quality of data by comparing the air temperature, air humidity, air pressure and cumulated rain with an official weather station of the hydro-meteorological network of the Canton Ticino in Switzerland, placed at the SUPSI Campus in Trevano. The prototype, which provides data in real-time, was installed near the reference station and was tested for a period of eight months. The time series were compared using the goodness-of-fit, scatter plots and density probability function in order to have a complete evaluation and overview of the performance of each sensor.

The first environmental parameter considered was the air temperature. The performance of the prototype is good, with a mean error of 0.26 °C and a standard deviation of 0.12 °C with daily averaged values. The residual’s trend increased from the summer to the winter season (Figure 8). This behavior can be explained by a different sensor’s responses to environmental climatic conditions like those identified in other studies [36], but this will not be possible to exclude a sensor drift until an analysis of a longer time series is performed. Meanwhile, the air relative humidity was evaluated, and showed more limits than the air temperature, even if the general accuracy was acceptable and compliant for the purpose of the project. After comparing the daily averaged time series of the two stations, the mean error is −1.60 %, with a standard deviation of 2.90. The sensor works better with higher values of humidity and is dependent on the air temperature. When the temperature decreases, the residuals’ dispersion and the mean error increase (Figure 12). The last air parameter taken into consideration was the air pressure, which, together with the air temperature, is the parameter which has the greatest similarity to the reference station. Also, in this case, the accuracy decreased with the temperature decrease, but the general behavior was good since the R-squared was 0.99 and the mean error was 0.38 hPa, with a standard deviation of 0.16 on the basis of the daily mean value series. The rain, which is one of the most important parameters for flood mitigation, was observed with good quality. In fact, the same events were detected (two were snow events) by the two stations, and very similar quantities were recorded. Nevertheless, the discrepancy with the reference station regarding the increase of the rain intensity must be evaluated in-depth, in order for the system to be effectively adopted as a proven alternative in tropical regions characterized by heavy short rain events.

In view of the presented results, the proposed solution can be considered a valuable alternative to conventional weather systems in helping fill the gap in the monitoring system for low-income and developing countries. The evaluated parameters showed very promising results, in particular for the air temperature and air pressure. Although the air humidity accuracy did not reach the same high level results as the other air parameters, considering the general complexity and uncertainty in measuring such environmental aspects, its quality is good enough to fulfill both the identified requirements and the objective of the project. Finally, the adopted rain gauge is very promising and can certainly be adopted in temperate regions, but further evaluations of its performance during heavy rain events must be carried out.

With this research we have proven that it is possible to implement a fully open weather monitoring solution based on open hardware, open software, open standard and open data, one that is capable of providing high quality data. Further activities will test this solution in a real-case experiment, in order to evaluate the system’s sustainability and durability in real conditions. Monthly reports of the monitored weather parameters will be automatically produced and shared with an open license through the CKAN portal thanks to the API interface. The reports will present daily minimum, maximum and mean values for each parameter, and will also include graphs of the time series.

## Figures and Tables

**Figure 1 sensors-19-01185-f001:**
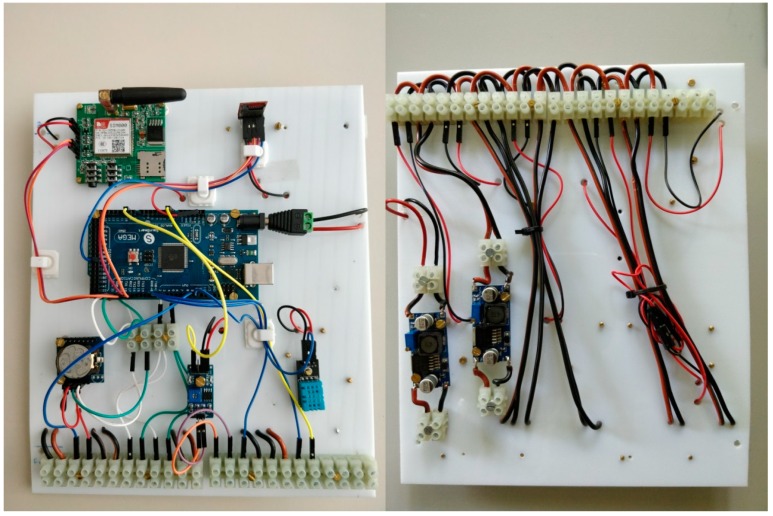
Front and back side of the weather station The Plexiglas board which is placed inside the main box container.

**Figure 2 sensors-19-01185-f002:**
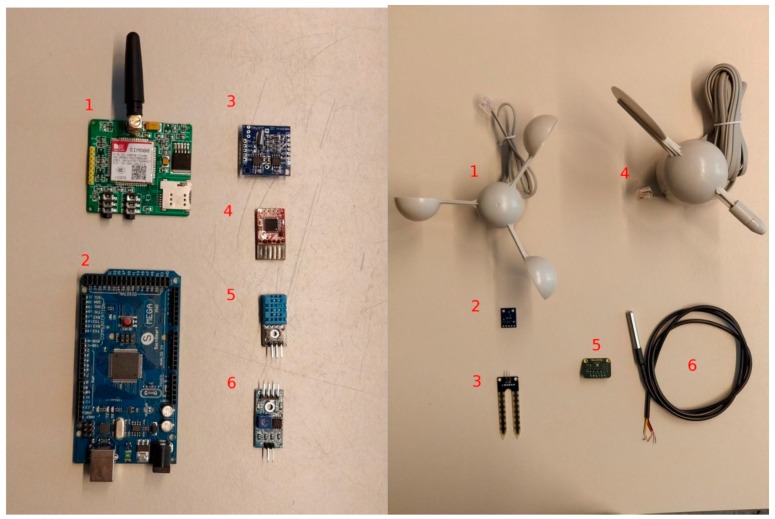
The components inside the box on the left, and outside of the box on the right (rain gauge is not present).

**Figure 3 sensors-19-01185-f003:**
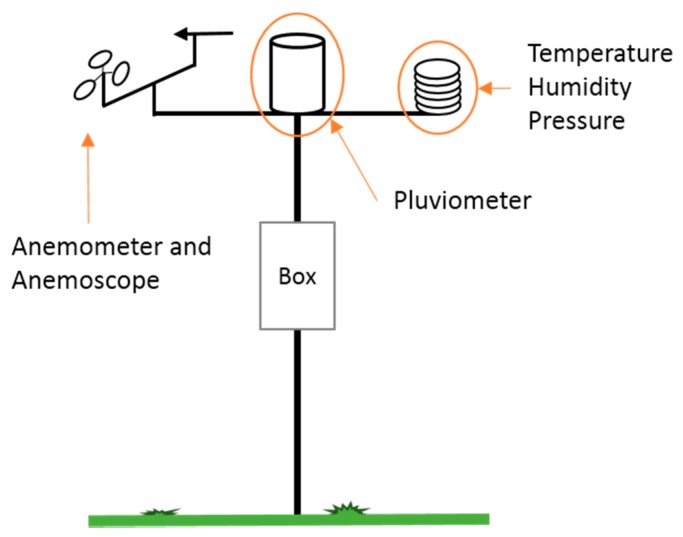
Weather station general schema.

**Figure 4 sensors-19-01185-f004:**
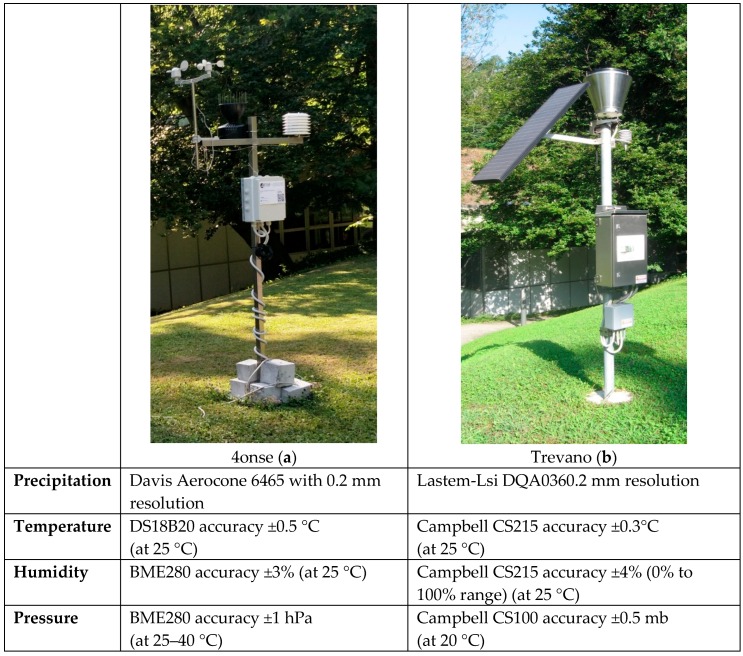
(**a**) Trevano sensors accuracy; and (**b**) 4onse-mod sensors accuracy.

**Figure 5 sensors-19-01185-f005:**
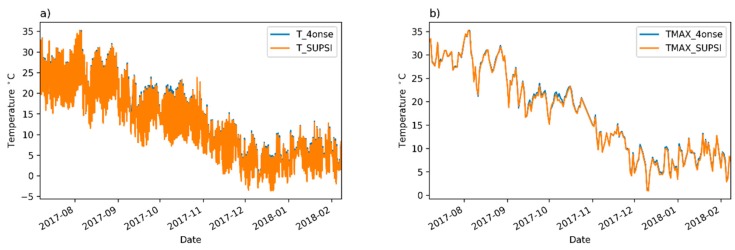
Temperature time series comparison: (**a**) 10 min aggregated values; (**b**) daily maximum values; (**c**) daily minimum values; (**d**) daily mean values.

**Figure 6 sensors-19-01185-f006:**
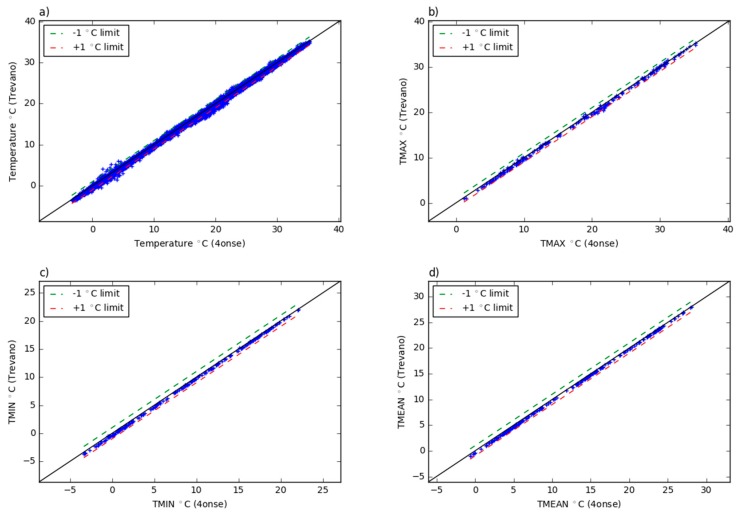
Trevano versus 4onse weather station temperature data: (**a**) 10 min aggregated values; (**b**) daily maximum values; (**c**) daily minimum values; (**d**) daily mean values.

**Figure 7 sensors-19-01185-f007:**
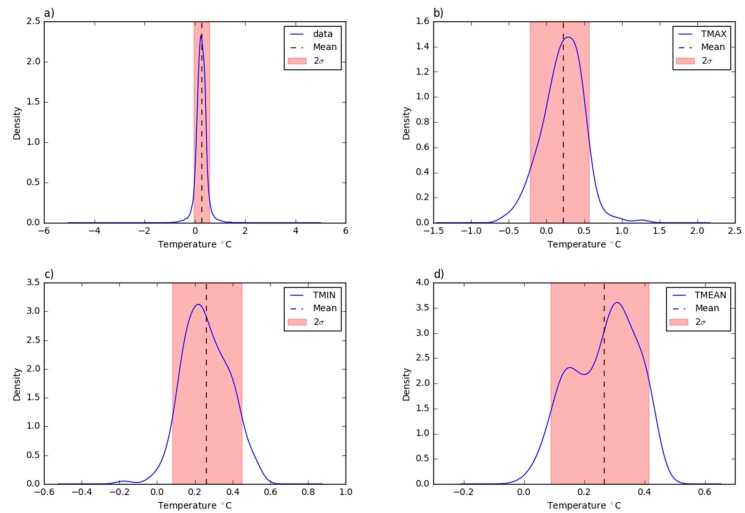
The probability density function of the residual deviations from the Trevano temperature time series: (**a**) 10 min aggregated values; (**b**) daily maximum values; (**c**) daily minimum values; (**d**) daily mean values.

**Figure 8 sensors-19-01185-f008:**
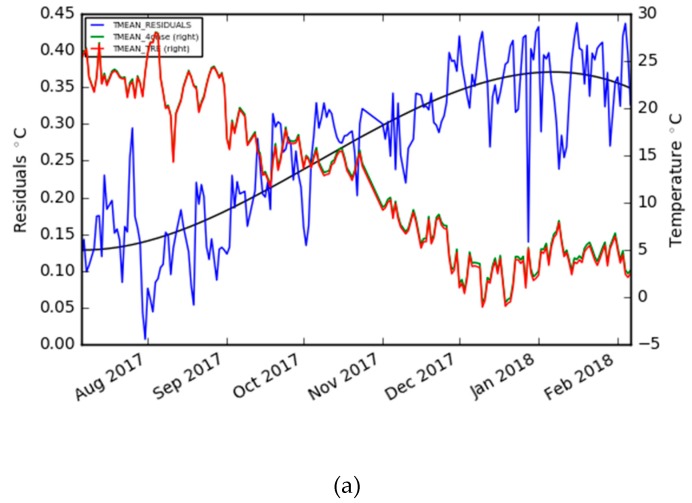
**(a**) The daily mean temperature of the Trevano and 4onse-mod stations plus residuals time; and (**b**) the residuals daily mean vs. daily mean temperature.

**Figure 9 sensors-19-01185-f009:**
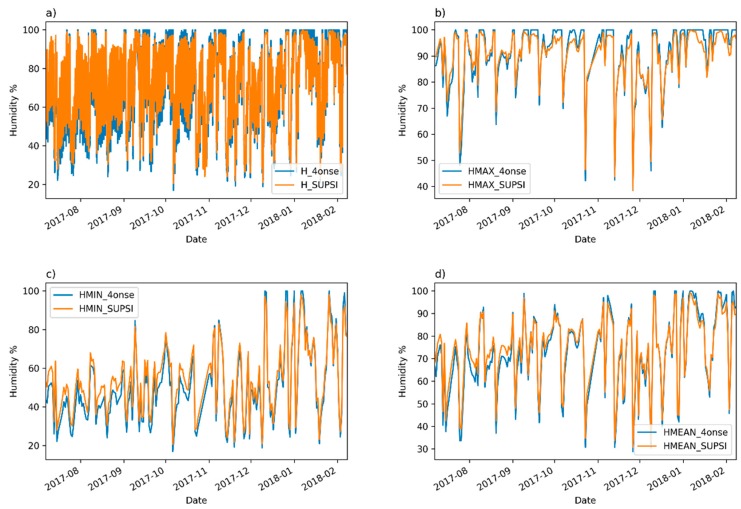
A humidity time series comparison: (**a**) 10 min aggregated values; (**b**) daily maximum; (**c**) daily minimum; (**d**) daily mean values.

**Figure 10 sensors-19-01185-f010:**
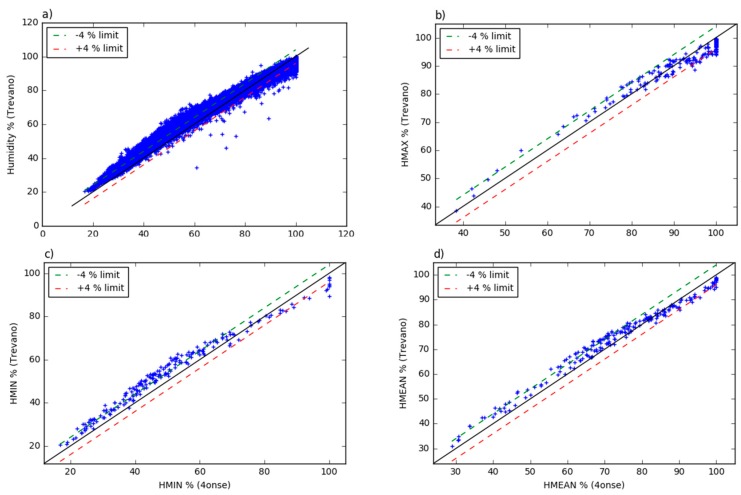
The Trevano versus 4onse weather station humidity data: (**a**) 10 min aggregated values; (**b**) daily maximum values; (**c**) daily minimum values; (**d**) daily mean values.

**Figure 11 sensors-19-01185-f011:**
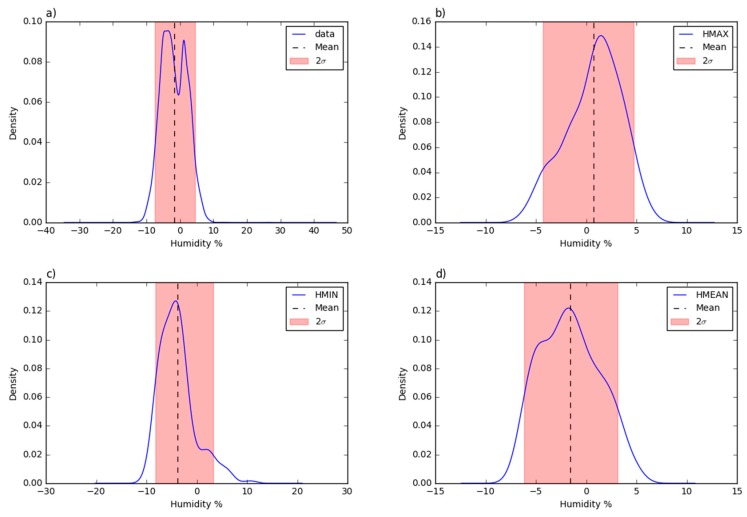
The probability density function of the residual deviations from the Trevano humidity time series: (**a**) 10 min aggregated values; (**b**) daily maximum values; (**c**) daily minimum values; (**d**) daily minimum values).

**Figure 12 sensors-19-01185-f012:**
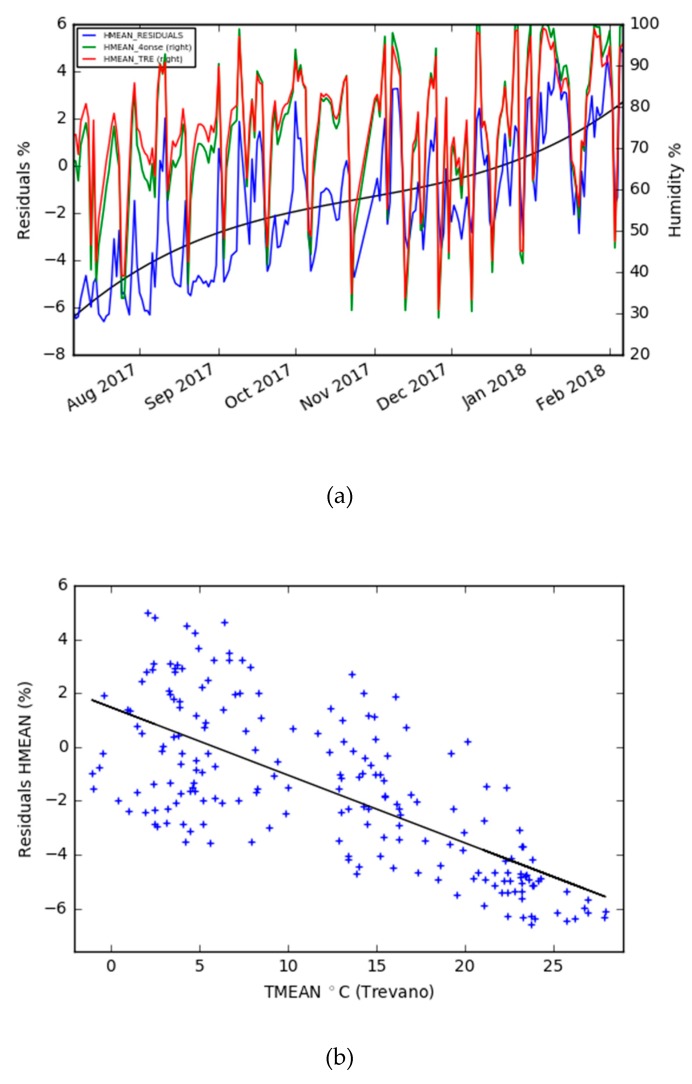
**(a**) The daily mean humidity of the Trevano and 4onse-mod stations plus the residuals time; and (**b**) the residuals of the daily mean humidity vs. daily mean temperature.

**Figure 13 sensors-19-01185-f013:**
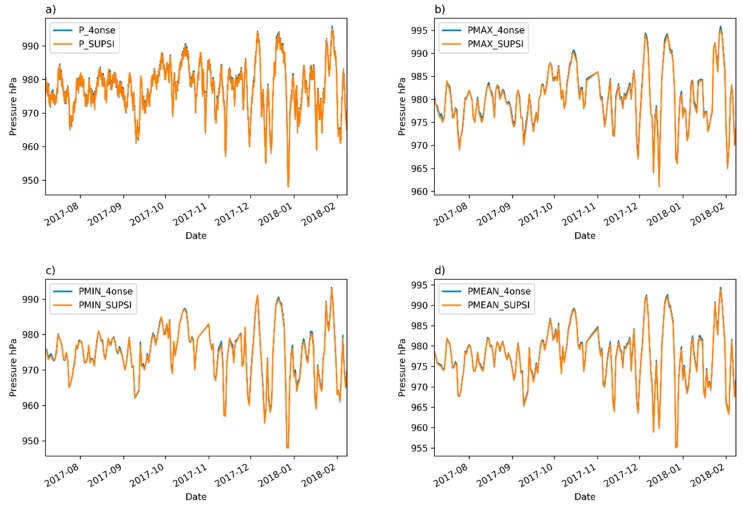
A pressure time series comparison: (**a**) 10 min aggregated values; (**b**) daily maximum values; (**c**) daily minimum values; (**d**) daily mean values.

**Figure 14 sensors-19-01185-f014:**
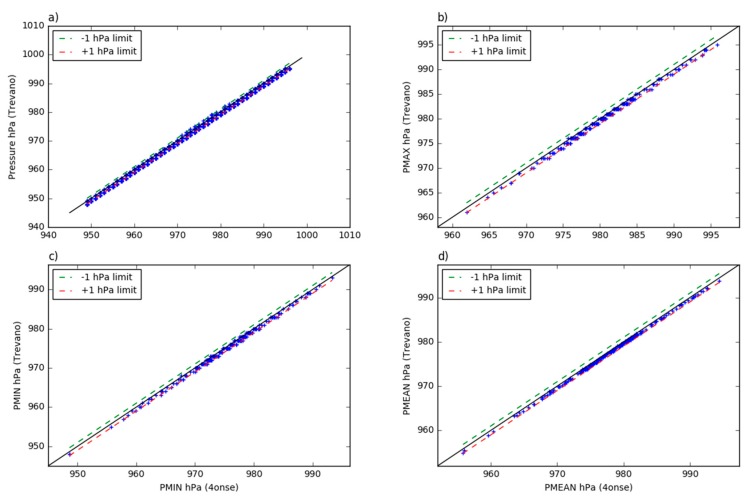
The Trevano versus 4onse weather station pressure data: (**a**) 10 min aggregated values; (**b**) daily maximum values; (**c**) daily minimum values; (**d**) daily minimum values.

**Figure 15 sensors-19-01185-f015:**
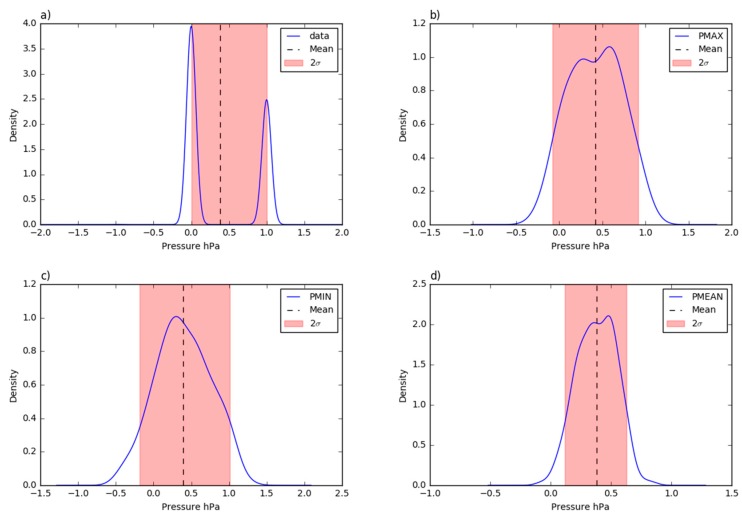
The probability density function of the residual deviations from the Trevano pressure time series: (**a**) 10 min aggregated values; (**b**) daily maximum values; (**c**) daily minimum values; (**d**) daily mean values.

**Figure 16 sensors-19-01185-f016:**
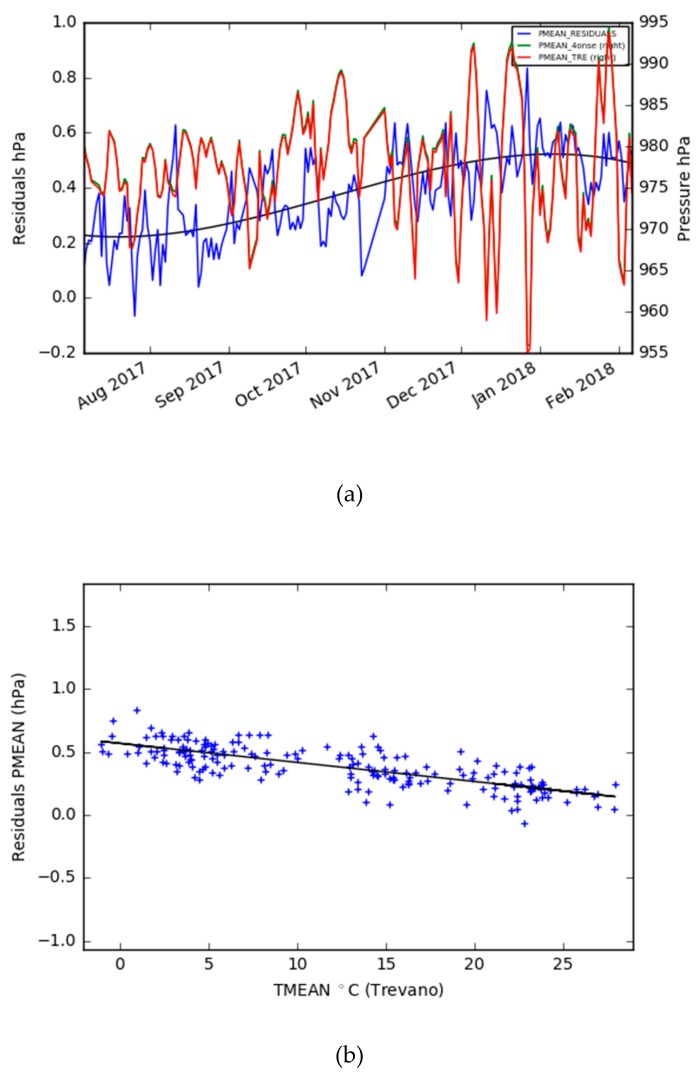
**(a**) The daily mean pressure of the Trevano and 4onse-mod stations plus the residuals time series; and (**b**) the residuals vs. the daily mean temperature.

**Figure 17 sensors-19-01185-f017:**
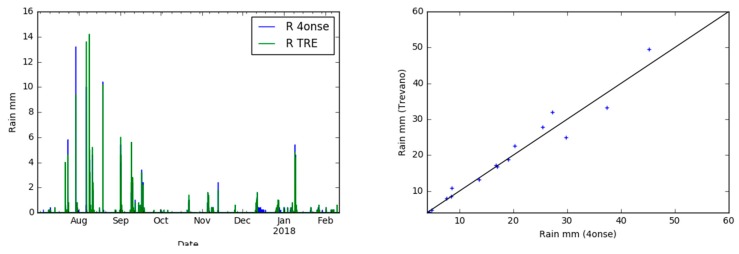
Rainfall time series graph (**left**) and scatter plot (4onse vs Trevano) (**right**).

**Table 1 sensors-19-01185-t001:** List of the components outside the box container.

Name	Description
DS18B20 (6)	The external waterproof sensor used to measure the air temperature. It reports temperatures ranging between −55 °C to 125 °C with an accuracy of +/−0.5 °C and a resolution of 0.0625 °C.
BME280 (5)	It measures air pressure and humidity with an accuracy of ±1 hPa and ±3% and resolution of 0.18 Pa and 0.008%, respectively.
Davis AeroCone 6465	The rain gauge has a resolution of 0.2 mm and a collection area of 214 cm^2^.
Anemometer and anemoscope (1,4)	This sensor is composed of a wind direction and speed plastic sensors.
YL-69 (3)	The soil moisture sensor can read the amount of water present in the soil surrounding it.
BH1750 (2)	This is a digital light sensor which provides lux with high resolution values.

**Table 2 sensors-19-01185-t002:** The square of the correlation coefficient between 4onse and the Trevano temperature series.

Temperature Time Series	R2	Mean Error	Std Error
10 min aggregated series	0.99	0.26	0.22
Daily max series	0.99	0.22	0.27
Daily mean series	0.99	0.26	0.10
Daily min series	0.99	0.26	0.12

**Table 3 sensors-19-01185-t003:** The square of the correlation coefficient between the 4onse and the Trevano humidity series.

Humidity Time Series	R2	Mean Error	Std Error
10 min aggregated series	0.99	−1.64	3.84
Daily max series	0.98	0.76	2.72
Daily mean series	0.99	−1.60	2.90
Daily min series	0.99	−3.87	3.58

**Table 4 sensors-19-01185-t004:** The square of the correlation coefficient between the 4onse and the Trevano pressure series.

Pressure Time Series	R2	Mean Error	Std Error
10 min aggregated series	0.99	0.38	0.48
Daily max series	0.99	0.41	0.31
Daily mean series	0.99	0.38	0.16
Daily min series	0.99	0.39	0.36

**Table 5 sensors-19-01185-t005:** Rain discretized events for the reference period.

Event n°	4onse (mm)	Trevano (mm)	Δ (mm)	Error (%)
1	7.6	8.0	−0.4	−5.0
2	13.6	13.2	+0.4	3.0
3	29.8	25.0	+4.8	19.2
4	27.2	32.0	−0.2	−0.6
5	8.6	10.8	−2.2	−20.4
6	20.2	22.6	−2.4	−10.6
7	45.2	49.6	−4.4	−8.9
8	25.4	27.8	−2.4	−8.6
9	4.8	4.8	0.0	0.0
10	4.4	4.2	+0.2	4.8
11	16.8	17.2	−0.4	−2.3
12	8.4	8.6	−0.2	−2.3
13	37.4	33.2	+4.2	12.7
14	19.0	18.8	+0.2	1.1
Mean error			−0.2	−1.3
Std. dev. error			2.4	9.8

**Table 6 sensors-19-01185-t006:** Comparison of the 1 h and 24 h rain max.

Rain Max	4onse (mm)	Trevano (mm)	Δ (mm)
24 h	29.8	24.8	+5
1 h	66.6	74.6	−8

**Table 7 sensors-19-01185-t007:** Total amount of rain measured (6 months).

4onse Total Rain (mm)	Trevano Total Rain (mm)
632.4	672.8

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
