# Peer review of "Boosting a Weather Monitoring System in Low Income Economies Using Open and Non-Conventional Systems: Data Quality Analysis"

_sensors, 2019, doi:10.3390/s19051185_

Round 1

Reviewer 1 Report

It was really interesting reading the paper.

I think that the results are promising and that the "4onse" architecture can be relevant not only for developing countries but also to increase the density of sensors in the developed countries.

Results in terms of accuracy are good enough and confirm that "4onse" can be used for different applications which do not require incredible accuracy in the measurements. 

I have only a couple of question/suggestions:

You have adopted a Davis AeroCone for collecting rainfall. Do you think that it will be possible to 3d print such a component to get a even more "Open Hardware" "4onse" architecture? In case it is feasible why do not write something about this in the manuscript?

I think you forget to comment table 7. However It is clear that the data are confirming the good performance of the architecture. Please ad some lines on this in to the text.

Some figures (e.g fig. 4 and 8, 12, 16)  are too much small. I really suggest to increase their sizes in order to improve their readability.

Author Response

Point 1: You have adopted a Davis AeroCone for collecting rainfall. Do you think that it will be possible to 3d print such a component to get a even more "Open Hardware" "4onse" architecture? In case it is feasible why do not write something about this in the manuscript?

Reponse 1: Thanks for this comment that underlines one aspect we are currently work on. Actually, we implemented a 3D printed rain gauge which is under testing. At this state the results are very promising, but we did not include a robust analysis since more data are needed. However, as you suggested, we added a small description on the work that we are doing on it. (line 239)

Point 2: I think you forget to comment table 7. However It is clear that the data are confirming the good performance of the architecture. Please ad some lines on this in to the text.

Response 2: The comment is correct, thanks for the note. We added a comment on the data showed in table 7. (line 404)

Point 3: Some figures (e.g fig. 4 and 8, 12, 16)  are too much small. I really suggest to increase their sizes in order to improve their readability.

Reponse 3: As you suggested the figures 4, 8, 12 and 16 size are increased and redefined.

Reviewer 2 Report

The work describes a system for environmental data collection and communication that is frugally developed and cheap for developing countries scenarios yet  is accurate, as claimed by the authors. And because of those points the system 4onse is claimed to be innovative, novel and unique. But it is hardly so as much more sophisticated systems with open source platforms with free software and hardware have been implemented in developing countries for areas like agriculture Sensors, geo-hazards sensor systems, industrial hazard monitoring and control, and more. Based on CKAN and other openly shared platforms with Arduino and mobile communications there have been many projects in developing and under developed countries. Many institutions in such countries are coming up with solutions like these everyday. They are now just applications based projects.

This paper basically compares the system 4onse with a scientific station and gives statistics on the various parameters and errors. This comparison becomes necessary to prove the validity of the study and nothing more.

The paper needs to be checked thoroughly and corrected in many places for English.

The literature review is weak, needs more detailed scrutiny.

Hence based on above points I recommend thorough revision of the paper with full implementation details on CKAN and availability to public and its effect. A study on that would be worthy of publication. At present it is just a technical report with bare skeletal research contribution.

Author Response

Point 1: The work describes a system for environmental data collection and communication that is frugally developed and cheap for developing countries scenarios yet  is accurate, as claimed by the authors. And because of those points the system 4onse is claimed to be innovative, novel and unique. But it is hardly so as much more sophisticated systems with open source platforms with free software and hardware have been implemented in developing countries for areas like agriculture Sensors, geo-hazards sensor systems, industrial hazard monitoring and control, and more. Based on CKAN and other openly shared platforms with Arduino and mobile communications there have been many projects in developing and under developed countries. Many institutions in such countries are coming up with solutions like these everyday. They are now just applications based projects.

Response 1: Thanks for the comment, we recognize several works with open hardware sensor, but in literature there's few references at our knowledge that report a system fully based on open technologies, standards and software with respect to climate observations. The prototype developed aims at maximizing the reproducibility and maintainability  using only components widely available in markets at global scale. This research presents a partial result of a project that intends to validate the technology in a real case scenario in Sri Lanka at watershed level. Hence, the effective sustainability will be verified to understand if this kind of system can effectively be an alternative to more expensive and conventional solutions. It is clear that this first analysis presented in this paper, rarely found in literature for such a kind of systems, is a pre-requisite for further tests at larger scale. The next steps of the project will give results on the durability and sustainability of such a kind of solution on not only the hardware part, but also on the software solutions chosen. However, we better explain our purpose in the manuscript based on your comment (line 31)

Point 2: The paper needs to be checked thoroughly and corrected in many places for English.

Reponse 2: We have carefully reviewed the language and applied several fixes / changes throughout the paper.

Point 3: The literature review is weak, needs more detailed scrutiny.

Reponse 3: We added more references that better frame the presented research. (references n°1,2,3,4,5,6,13,19,29)

Point 4: Hence based on above points I recommend thorough revision of the paper with full implementation details on CKAN and availability to public and its effect. A study on that would be worthy of publication. At present it is just a technical report with bare skeletal research contribution.

Response 4: Thank you for this comment, we added details on the future developments on this (line 475). However, at this state, we would like just to make a quality validation of the data observed by the station. In future also this part which is still in development will be taken into consideration for a publication.

Round 2

Reviewer 2 Report

Much improved version now. Addresses the majority of questions I raised. No more comments from my side.